# Matrix Multiplicative Weights Updates in Quantum Zero-Sum Games: Conservation Laws & Recurrence

**Rahul Jain**
NUS
`rahul@comp.nus.edu.sg`

**Georgios Piliouras**
SUTD
`georgios@sutd.edu.sg`

**Ryann Sim**
SUTD
`ryann_sim@mymail.sutd.edu.sg`

## Abstract

Recent advances in quantum computing and in particular, the introduction of quantum GANs, have led to increased interest in quantum zero-sum game theory, extending the scope of learning algorithms for classical games into the quantum realm. In this paper, we focus on learning in quantum zero-sum games under *Matrix Multiplicative Weights Update* (a generalization of the multiplicative weights update method) and its continuous analogue, *Quantum Replicator Dynamics*. When each player selects their state according to quantum replicator dynamics, we show that the system exhibits conservation laws in a quantum-information theoretic sense. Moreover, we show that the system exhibits Poincaré recurrence, meaning that almost all orbits return arbitrarily close to their initial conditions infinitely often. Our analysis generalizes previous results in the case of classical games [48, 42].

## 1 Introduction

The Nash equilibrium has always been a central concept in non-cooperative game theory. While the existence of Nash equilibria is well known in finite games where mixed strategies are allowed [44], it is less clear how to efficiently compute these equilibria themselves. Indeed, much work revolves around finding methods to solve for Nash equilibria in different contexts, as well as analyzing the complexity of solving for the Nash [33, 15].

Despite the dominance of equilibrium computation in classical theory, recent works have begun to move towards attempting to understand the nature of learning in games [23, 51]. The family of regret-minimizing algorithms known as Follow-The-Regularized-Leader (FTRL) have been studied extensively, both for discrete time [7, 13, 45, 49] and continuous time [42, 21, 48, 47] dynamics. These algorithms have found many applications in the domain of machine learning - one such example is that of Generative Adversarial Networks (GANs) [25], which have been successfully modeled using zero-sum game theory. This rich connection has led to a stronger understanding of adversarial learning and improved practical performance of GANs in various settings [21, 32, 36].

On the other side of the coin, quantum computing is a field which has garnered much interest in the machine learning community. Many quantum machine learning algorithms have been proposed, extending standard classical algorithms. One such example is quantum GANs [14, 39, 12]. While such works describe the architecture of quantum GANs, we still do not fully understand the behavior of learning algorithms in the quantum setting. In this paper, we provide an initial analysis of learning in zero-sum quantum games. We do so by studying the learning behaviour of the ubiquitous Multiplicative Weights Update (MWU) update rule [37, 22, 5], a specific instance of the FTRL framework, when applied to such games.

36th Conference on Neural Information Processing Systems (NeurIPS 2022).

There have been several important results studying MWU and its continuous counterpart, *replicator dynamics*. [7] showed that classical MWU does not converge in day-to-day behaviour to the mixed Nash equilibrium when applied to two player zero-sum games. [42] showed that the orbits of players in zero-sum games exhibit Poincaré recurrence when using replicator dynamics. However, quantum systems can oftentimes behave in radically different ways to their classical counterparts. Indeed, allowing for quantum strategies can create situations wherein players can gain greater payoffs than if they were playing using only classical strategies (see e.g. the Mermin–Peres magic square game [40, 46]). As such, our extension of the classical results to the quantum realm has to be treated with care.

We focus on a matrix generalization of MWU known as *Matrix Multiplicative Weights Update* (MMWU), which changes the paradigm of standard MWU from cost vectors to cost matrices, and from probability vectors to density matrices. This generalization has been independently discovered and studied by [54] as Matrix Exponentiated Gradient Updates and [5] as the Matrix Multiplicative Weights algorithm. Applications of the MMWU algorithm include solving semi-definite programs (SDPs) [6] and obtaining bounds on the sample complexity for learning problems in quantum computing [30]. From a game theoretic standpoint, [28] analyzed the MMWU algorithm for quantum zero-sum games, proving time-average convergence to an approximate Nash equilibrium. We extend this result by studying the learning behaviour of MMWU in quantum zero-sum games, which could lay the groundwork for developing novel algorithms that achieve convergence to the Nash in day-to-day behaviour in quantum zero-sum games, similar to the case of their classical analogues [16, 41].

**Contributions.**   In this paper, we first study the dynamics of MMWU in quantum zero-sum games, utilizing tools from information theory and classical game theory (Section 3). We provide bounds on the rate of change of the total quantum relative entropy between a fully mixed Nash equilibrium and the evolving state of the system. We then study the continuous counterpart of MMWU, which we call quantum replicator dynamics (Section 4). We show that the aforementioned total quantum relative entropy is a constant of motion. Furthermore, the dynamics do not converge to equilibrium (in the day-to-day sense) but rather exhibit a weaker form of approximate periodicity known as *Poincaré recurrence*. Our proof of this result departs from the standard classical method, representing our main technical novelty. Finally, we present several simulations which corroborate our theoretical results (Section 5).

## 2   Preliminaries and Definitions

### 2.1   Quantum Theory

**Basic concepts.**   Similarly to [28] and in order to make the presentation self-contained, we will start by introducing some basic concepts of quantum information and game theory.

First, we refer to a quantum *register* as a collection of qubits representing a message that is transferred from one party to another. We associate a vector space $\mathcal{H} = \mathbb{C}^n$ with any quantum register. This intuitively represents the maximum number of distinct classical states that can be stored in the register without error. The *state* of a quantum register is represented by a *density matrix*, which is an $n \times n$ positive semi-definite matrix with trace 1. We will use $D(\mathcal{H})$ to denote the set of all density matrices associated with a register that is described by $\mathcal{H}$. One can naturally view such density matrices as linear operators acting on $\mathcal{H}$.

When two registers with associated spaces $\mathcal{A} = \mathbb{C}^n$ and $\mathcal{B} = \mathbb{C}^m$ are considered as a joint register, the associated space is the tensor product $\mathcal{A} \otimes \mathcal{B} = \mathbb{C}^{nm}$. If the two registers are independently prepared in states described by $\rho$ and $\sigma$, then the joint state is described by the $nm \times nm$ density matrix $\rho \otimes \sigma$.

Next, for a given vector space $\mathcal{H} = \mathbb{C}^n$, we define $\mathrm{L}(\mathcal{H})$ as the set of all $n \times n$ complex matrices. Furthermore, we denote the subset of $\mathrm{L}(\mathcal{H})$ given by *Hermitian* matrices as $\mathrm{Herm}(\mathcal{H})$. A Hermitian matrix $A$ satisfies the equality $A = A^\dagger$, where $A^\dagger$ denotes the *adjoint* (or conjugate transpose) of matrix $A$. Subsequently, we define $\mathrm{Pos}(\mathcal{H})$ as the subset of $\mathrm{Herm}(\mathcal{H})$ which consists of all positive semi-definite $n \times n$ matrices.

Finally, the *Hilbert-Schmidt inner product* on $\mathrm{L}(\mathcal{H})$ is defined as $\langle A, B \rangle = \mathrm{Tr}(A^\dagger B)$ for all $A, B \in \mathrm{L}(\mathcal{H})$. Note that $\langle A, B \rangle$ is a real number for any Hermitian matrices $A$ and $B$, and is also additionally non-negative if $A$ and $B$ are positive semi-definite.

**Measurements and observables.**   In the context of game theory, we are also interested in the concepts of quantum *measurements* and *observables*. An observable is simply a property of the quantum system which is measurable. The measurement of a register having associated vector space $\mathcal{H} = \mathbb{C}^n$ is a collection of linear operators $\{P_i : 1 \leq i \leq k\} \subset \mathrm{Pos}(\mathcal{H})$ which satisfies $\sum_{i=1}^k P_i = \mathbb{1}_{\mathcal{H}}$, where $\mathbb{1}_{\mathcal{H}}$ is the identity matrix on $\mathcal{H}$. If the register corresponding to $\mathcal{H}$ is in a state defined by density matrix $\rho$ and the measurement described by $P_i$ is performed, each outcome $i$ will be observed with probability $\langle P_i, \rho \rangle$. An important note is that two mixed states with the same density matrix are indistinguishable from each other by any measurement.

**Additional notation and definitions.**   A linear mapping of the form $\Phi : \mathrm{L}(\mathcal{B}) \to \mathrm{L}(\mathcal{A})$ is called a super-operator. The adjoint super-operator to $\Phi$ is $\Phi^\dagger : \mathrm{L}(\mathcal{A}) \to \mathrm{L}(\mathcal{B})$ and is uniquely determined by the condition:

$$\langle A, \Phi(B) \rangle = \langle \Phi^\dagger(A), B \rangle$$

A super-operator $\Phi : \mathrm{L}(\mathcal{B}) \to \mathrm{L}(\mathcal{A})$ is said to be positive if $\Phi(P)$ is positive semi-definite for every choice of positive semi-definite operator $P \in \mathrm{Pos}(\mathcal{B})$. In addition, $\Phi^\dagger$ is positive if and only if $\Phi$ is positive. There is a one-to-one and linear correspondence between the collection of operators of the form $R \in \mathrm{L}(\mathcal{A} \otimes \mathcal{B})$ and the collection of super-operators $\Phi : \mathrm{L}(\mathcal{B}) \to \mathrm{L}(\mathcal{A})$ defined above. Specifically, for each super-operator $\Phi$ we can define an operator $R$ as follows:

$$R = \sum_{1 \leq i, j \leq m} \Phi(E_{i,j}) \otimes E_{i,j} \tag{1}$$

where $E_{i,j}$ is the matrix with a 1 in entry $(i, j)$ and 0 elsewhere. Conversely, given an operator $R \in \mathrm{L}(\mathcal{A} \otimes \mathcal{B})$, we can define:

$$\Phi(B) = \mathrm{Tr}_{\mathcal{B}}(R(\mathbb{1}_{\mathcal{A}} \otimes B^\top)) \tag{2}$$

This correspondence is linear and one can translate back and forth between the two as needed for an given application. $R$ is assumed to be positive semi-definite, so the corresponding super-operator $\Phi$ is also positive.

## 2.2   Quantum Game Theory

A quantum system which can be manipulated by any number of agents, and where the utility of the moves is well defined, quantified and ordered can be conceived as a quantum game. In particular, a 2-player zero-sum quantum game sees players Alice and Bob each sending a quantum state to a referee, who then performs a measurement on these two states to determine their payoffs. Let $\mathcal{A} = \mathbb{C}^n$ and $\mathcal{B} = \mathbb{C}^m$ be the vector spaces that correspond to the state that Alice and Bob send to the referee.

In order to determine the payoffs of the players' actions, the referee performs a joint measurement where Alice's and Bob's states are viewed as a single register. Thus, the referee's measurement can be described by a collection

$$\{R_a : a \in \mathcal{S}\} \subset \mathrm{Pos}(\mathcal{A} \otimes \mathcal{B}) \tag{3}$$

which satisfies the condition $\sum_{a=1}^k R_a = \mathbb{1}_{\mathcal{A} \otimes \mathcal{B}}$. We associate each possible measurement outcome $a$ with a payoff for each player. Since we are only considering zero-sum games, if the payoff that Alice receives from the referee is $v(a)$, then Bob's corresponding payoff will be $-v(a)$. Henceforth, we refer to the states sent by Alice and Bob to the referee as $\rho$ and $\sigma$ respectively. For a given choice of $\rho$ and $\sigma$, Alice's expected payoff is:

$$u(\rho, \sigma) = \sum_{a=1}^k v(a) \langle R_a, \rho \otimes \sigma \rangle = \langle R, \rho \otimes \sigma \rangle \tag{4}$$

where $R = \sum_{a=1}^k v(a) R_a$. Likewise, Bob's corresponding expected payoff is $-u(\rho, \sigma) = -\langle R, \rho \otimes \sigma \rangle$. $R$ is referred to throughout the rest of the paper as a *payoff observable*. A necessary and sufficient

condition for matrix $R$ acting on $\mathcal{A} \otimes \mathcal{B}$ to be obtained from some real valued payoff function $v$ is that $R$ is Hermitian. From Equation 2, we can equivalently define the expected payoff of Alice as $u(\rho, \sigma) = \langle \rho, \Phi(\sigma) \rangle$. We will use the latter formulation for the remainder of the paper.

A key notion of equilibrium in classical game theory is the Nash equilibrium. In the quantum setting, we define the pair of quantum states $(\rho^*, \sigma^*)$ as a Nash equilibrium of $R$ if

$$u(\rho^*, \sigma^*) \geq u(\rho, \sigma^*) \quad \text{and} \quad u(\rho^*, \sigma^*) \geq u(\rho^*, \sigma) \tag{5}$$

for all $\rho, \sigma$. That is, neither Alice nor Bob would prefer to unilaterally deviate from playing $\rho^*$ and $\sigma^*$ respectively. On another note, since the set of available strategies (density matrices) for both agents is convex and compact and the utility function of Alice is bilinear, standard extensions of the of von Neumann's Min-Max Theorem apply [20].

Finally, we define the notion of a 'fully mixed' Nash equilibrium. An equilibrium $(\rho^*, \sigma^*)$ is fully mixed if $\rho^*, \sigma^*$ are full rank.

### 2.3 Information Theory

We also introduce several information theoretic concepts which will be referenced throughout the paper.

**Shannon entropy.** The Shannon entropy of a random variable $X$ where each strategy $x$ is obtained with probability $p(x)$ is given by $H(X) = -\sum_x p(x) \log p(x)$. This intuitively is a measure of randomness or uncertainty in the system. A natural generalization of the Shannon entropy to a quantum context is the von Neumann entropy. For a quantum mechanical system defined by density matrix $\rho$, the von Neumann entropy is given by $S(\rho) = -\operatorname{Tr}(\rho \log \rho)$.

**Bregman divergence.** We are also interested in the notion of Bregman divergence, which measures the distance between two points. Let $x^* \in \mathcal{X}$ be a Nash equilibrium and let $x \in \mathcal{X}$ be an arbitrary strategy profile. Also, let F be a continuously-differentiable, strictly convex function. The Bregman divergence from $x^*$ to $x$ is given by $D_F(x^* \| x) = F(x^*) - F(x) - \langle \nabla F(x), x^* - x \rangle$.

In the context of quantum games, the Bregman divergence is defined using matrix notation. We define the quantum relative entropy between two quantum states using the von Neumann entropy $S(\rho)$. In particular, the quantum relative entropy between two quantum states $\rho$ and $\sigma$ is defined as $S(\rho \| \sigma) = \operatorname{Tr}(\rho(\log \rho - \log \sigma))$.

### 2.4 Dynamical Systems

**Flows.** Consider a differential equation $\dot{p} = f(p)$ on a topological space $\mathcal{P}$. The existence and uniqueness theorem for ordinary differential equations guarantees that we can write the unique solution to the differential equation as a continuous map $\phi : \mathcal{P} \times \mathcal{H} \to \mathcal{P}$. This is referred to as the *flow of the differential equation* such that for any point $p \in \mathcal{P}$, $\phi(p, -)$ defines a function of time corresponding to the trajectory of $p$. Conversely, fixing a time $t$ provides a map $\phi^t \equiv \phi(-, t) : \mathcal{P} \to \mathcal{P}$. In Section 4, we introduce the notion of quantum replicator dynamics, which are Lipschitz continuous differential equations. Hence, a unique flow $\phi$ of these replicator dynamics exists.

**Liouville's theorem.** Liouville's theorem can be applied to any system of ordinary differential equations with a continuously differentiable vector field $\xi$ on an open domain $\mathcal{Y} \in \mathbb{R}^d$. The divergence of $\xi$ at $y \in \mathcal{Y}$ is the trace of the Jacobian at $y$: $\operatorname{div} \xi(y) = \sum_{i=1}^d \frac{\partial \xi_i}{\partial y_i}(y)$. Because the divergence is continuous, it is integrable on Lebesgue measurable subsets of $\mathcal{Y}$. Given any such set C, define the image of C under flow $\phi$ at time $t$ as $C(t) = \{\phi(c, t) : c \in C\}$. $C(t)$ is measurable and of volume $\operatorname{vol}[C(t)] = \int_{C(t)} d\mu$. Liouville's formula states that the time derivative of the volume $\operatorname{vol}[C(t)]$ exists and links it to the divergence of $\xi$:

$$\frac{d}{dt}[\operatorname{vol} C(t)] = \int_{C(t)} \operatorname{div}(\xi) d\mu \tag{6}$$

If $\operatorname{div} \xi(y)$ is null at any $y \in \mathcal{Y}$, then the volume is conserved. Since $\operatorname{div} \xi$ is continuous, the converse statement is also true - if the volume is conserved on any open set, $\operatorname{div} \xi(y)$ has to be null at any point $y \in \mathcal{Y}$.

---

**Algorithm 1:** Parallel Approximation of Equilibrium Point

---

Let $\mu = \epsilon/8$ and let $N = \lceil 64 \log (nm)/\epsilon^2 \rceil$.
**Initialize**: $A_0 = \mathbb{1}_{\mathcal{A}}$, $\rho_0 = A_0/\text{Tr}(A_0)$, $B_0 = \mathbb{1}_{\mathcal{B}}$, and $\sigma_0 = B_0/\text{Tr}(B_0)$.
**for** $j = 1 \ldots N$ **do**
$\quad A_j = \exp\left(\mu \sum_{i=0}^{j-1} \Phi(\sigma_i)\right)$
$\quad \rho_j = A_j/\text{Tr}(A_j)$
$\quad B_j = \exp\left(-\mu \sum_{i=0}^{j-1} \Phi^*(\rho_i)\right)$
$\quad \sigma_j = B_j/\text{Tr}(B_j)$
**end for**

---

**Diffeomorphisms of flows.** A function **f** between two topological spaces is called a diffeomorphism if i) **f** is a bijection, ii) **f** is continuously differentiable, iii) **f** has a continuously differentiable inverse. Two flows $\Phi^t : A \rightarrow A$ and $\Psi^t : B \rightarrow B$ are diffeomorphic if there exists a diffeomorphism $\mathbf{f} : A \rightarrow B$ such that for each $x \in A$ and $t \in \mathbb{R}$, $\mathbf{f}(\Phi^t(p)) = \Psi^t(\mathbf{f}(p))$. For the purpose of our analysis, the replicator dynamics defined in Equations 17 are translated via a diffeomorphism from the interior of $\mathcal{P}$ to a space $\mathcal{C} = \Pi_{i \in V} \mathbb{R}^{n-1}$, which allows us to show certain desirable properties.

**Poincaré recurrence.** The concept of Poincaré recurrence arises from Henri Poincaré's celebrated 1890 work regarding the three body problem [50]. He proved that if a dynamical system preserves volume and always remains bounded in its orbits, almost all trajectories return arbitrarily close to their initial position, and do so infinitely often.

**Theorem 2.1** (Poincaré recurrence). *If a flow preserves volume and has only bounded orbits then for each open set there exist orbits that intersect the set infinitely often.*

## 3 MMWU in Quantum Zero-Sum Games

In [28], the MMWU algorithm for zero-sum games is shown to exhibit time-average convergence to an approximate Nash equilibrium in two-player quantum zero-sum games. The MMWU algorithm is shown in Algorithm 1.

Note that here we focus specifically on two-player games, and utilize the expected payoffs defined via super-operators $\Phi : \text{L}(\mathcal{A}) \rightarrow \text{L}(\mathcal{B})$ as seen in Equation 2. Moreover, $\mu$ is the step-size in the quantum algorithm.

In this section, we examine closely the update steps for each player in the MMWU algorithm (Algorithm 1) and analyze the limiting behaviour of the total quantum relative entropy in the system. First, we introduce two useful facts which will aid in the analysis.

*Fact* 3.1 (Golden-Thompson inequality [24, 53]). Let $A, B$ be Hermitian matrices. Then

$$\text{Tr} \exp(A + B) \leq \text{Tr} \exp(A) \exp(B) \tag{7}$$

*Fact* 3.2. Let $0 \leq A \leq \mathbb{1}$ be a PSD matrix and $\delta$ be a real number. Then,

$$\exp(\delta A) \leq \mathbb{1} + \delta \exp(\delta) A \tag{8}$$

Next, we put forward two corollaries which will help us prove Theorem 3.5. We first define $\Delta S(\rho^* \| \rho_j) = S(\rho^* \| \rho_j) - S(\rho^* \| \rho_{j-1})$ and $\Delta S(\sigma^* \| \sigma_j) = S(\sigma^* \| \sigma_j) - S(\sigma^* \| \sigma_{j-1})$.

**Corollary 3.3.** *The change in the sum of quantum relative entropies in a quantum zero-sum game between a fully-mixed Nash equilibrium and the players' strategies is given by:*

$$\Delta S(\rho^* \| \rho_j) + \Delta S(\sigma^* \| \sigma_j) = \log \frac{\text{Tr} A_j}{\text{Tr} A_{j-1}} + \log \frac{\text{Tr} B_j}{\text{Tr} B_{j-1}} \tag{9}$$

**Corollary 3.4.** *The following trace inequalities hold for PSD matrices $A$ and $B$ updated with MMWU:*

$$\Delta S(\rho^* \| \rho_j) + \Delta S(\sigma^* \| \sigma_j) \geq \mu \exp(-\mu) \text{Tr}(\rho_j \Phi(\sigma_{j-1})) - \mu \exp(\mu) \text{Tr}(\rho_{j-1} \Phi(\sigma_j)) \tag{10}$$

$$\Delta S(\rho^* \| \rho_j) + \Delta S(\sigma^* \| \sigma_j) \leq \mu \exp(\mu) \text{Tr}(\rho_{j-1} \Phi(\sigma_{j-1})) - \mu \exp(-\mu) \text{Tr}(\rho_{j-1} \Phi(\sigma_{j-1})) \tag{11}$$

The proof of Corollary 3.4 relies on Facts 3.1 and 3.2, as well as the equality shown in Corollary 3.3.

In many practical scenarios, one would use decreasing step-sizes when running MMWU. As such, we utilize Corollary 3.4 and take the limit as step-size $\mu$ goes to $0$ in order to show the following theorem:

**Theorem 3.5.** *The change in the sum of quantum relative entropies between a fully mixed Nash equilibrium and the player's strategies and in a two-player zero-sum quantum game tends to zero when step-size $\mu \to 0$. Specifically,*

$$\lim_{\mu \to 0} \frac{1}{\mu} \left( \Delta S(\rho^* \| \rho_j) + \Delta S(\sigma^* \| \sigma_j) \right) = 0 \tag{12}$$

The proof of Theorem 3.5, along with Corollaries 3.3 and 3.4 can be found in Appendix C.

Theorem 3.5 further motivates an investigation into the continuous time variant of MMWU, which we call quantum/matrix replicator dynamics. In particular, we show that in the continuous case, the sum of quantum relative entropies is invariant. We introduce and study quantum replicator dynamics in detail in Section 4.

## 4 Replicator Dynamics in Quantum Zero-Sum Games

We have seen that in the case of discrete dynamics (MMWU), as the step-size becomes infinitesimal, the total quantum relative entropy between the Nash equilibrium and the system state tends to stabilize. A natural question would then be: does the same result hold in continuous time? In order to explore this question in greater detail, we first need to define the continuous analogue of MMWU, the *quantum replicator dynamics*, which has well known classical analogues [51].

We start by rewriting the MMWU update steps, but now defined over a continuous time interval $[0, t]$:

$$A(t) = \int_0^t \Phi(\sigma(\tau)) d\tau \tag{13}$$

$$\rho(t) = \exp(A(t)) / \mathrm{Tr}(\exp(A(t))) \tag{14}$$

$$B(t) = -\int_0^t \Phi^\dagger(\rho(\tau)) d\tau \tag{15}$$

$$\sigma(t) = \exp(B(t)) / \mathrm{Tr}(\exp(B(t))) \tag{16}$$

Note here that we shift the exponential terms from the definition of $A(t)$ and $B(t)$ to the corresponding $\rho(t)$ and $\sigma(t)$ terms. This will help simplify some of the proof techniques later on in the paper. Furthermore, in the rest of the paper we will typically drop from the notation the explicit dependence on $t$ to ease with the notational burden.

It is important to note the following observation, which will be helpful in our later analysis.

*Observation* 4.1. The discrete-time trajectories $\rho_j$ and $\sigma_j$ defined in Algorithm 1 are a standard Euler discretization (with step $\mu$) of the continuous-time trajectories $\rho(t)$ and $\sigma(t)$ defined in Equations 14 and 16.

We now define the *quantum replicator dynamics* as:

$$d\rho/dt = \frac{d}{dt} \left( \frac{\exp(A)}{\mathrm{Tr}(\exp(A))} \right), \quad d\sigma/dt = \frac{d}{dt} \left( \frac{\exp(B)}{\mathrm{Tr}(\exp(B))} \right) \tag{17}$$

It is worth noting that in the classical/commuting setting, one can write the replicator equations in a form that describes the relative utility that one agent obtains as compared to the average utility overall. However, in the quantum case this is not possible in general, since it relies on the assumption that $\int_0^t \Phi(\sigma(\tau)) d\tau$ and $\Phi(\sigma(t))$, and respectively $\int_0^t \Phi^\dagger(\rho(\tau)) d\tau$ and $\Phi^\dagger(\rho(t))$ commute.

As a consequence of Observation 4.1, we can also conclude that the dynamical system described by the replicator dynamics defined in Equations 17 is a limit case of the dynamical system described by the MMWU algorithm as $\mu \to 0$.

We are now able to state the main theorem for quantum relative entropy in quantum replicator dynamics.

**Theorem 4.2.** *When applying matrix/quantum replicator dynamics in a quantum zero-sum game with a fully-mixed Nash equilibrium $(\rho^*, \sigma^*)$, the sum of quantum relative entropies between the fully-mixed Nash equilibrium and the state of the system $(\rho(t), \sigma(t))$ is invariant on every system trajectory, i.e.:*

$$\frac{d\big(S(\rho^*\|\rho(t)) + S(\sigma^*\|\sigma(t))\big)}{dt} = 0 \tag{18}$$

## 4.1 Poincaré Recurrence in Quantum Zero-Sum Games

Now that we have described analytical results surrounding the day-to-day behaviour of quantum replicator dynamics, we seek to understand the *dynamics* of the trajectories. After all, invariance of quantum relative entropy does not fully describe how the system moves over time. We show that for any two-player zero-sum quantum game updated with replicator dynamics, the system exhibits *Poincaré recurrence*, insofar as the game is zero-sum and has a fully-mixed Nash equilibria. As introduced in Section 2, the notion of Poincaré recurrence is a weaker version of periodicity. To be precise, for almost all initial conditions $\rho_0 \in \mathcal{P}$, the replicator dynamics return arbitrarily close to $\rho_0$ infinitely often.

**Theorem 4.3.** *The quantum replicator dynamics given in Equations 17 are Poincaré recurrent in any two player zero-sum game which has a fully-mixed Nash equilibrium.*

The proof of this main theorem involves carefully piecing together several auxiliary results, which we will describe in the rest of the section. Furthermore, we stress that due to the non-commutative nature of quantum systems, the standard (classical) approach of differentiating the discrete-time dynamics in the primal space of probability distributions does not apply directly unless we have the highly unlikely situation where $\int_0^t \Phi(\sigma(\tau))d\tau$ and $\Phi(\sigma(t))$ (resp. $\int_0^t \Phi^\dagger(\rho(\tau))d\tau$ and $\Phi^\dagger(\rho(t))$) commute. This problem with carrying over the standard approach is explicitly discussed in Appendix D.

For the proof in the quantum setting, we first define a *canonical transformation* on the space of the matrices $A(t)$ and $B(t)$, which will be crucial in proving the theorem.

**Definition 4.4** (Canonical transformation). We define the canonical transformation of $A'(t)$ and $B'(t)$ to be a mapping of $A(t)$ and $B(t)$ as defined by Equations 13 and 15. In particular, we define

$$\begin{aligned}
A'(t) &= A(t) - (v^\dagger A(t)v)\mathbb{1} \\
B'(t) &= B(t) - (v^\dagger B(t)v)\mathbb{1}
\end{aligned} \tag{19}$$

where $v$ is a fixed vector defined as $v = [1, 0 \ldots 0]^\top$, such that the values of $v^\dagger A(t)v$ and $v^\dagger B(t)v$ are real numbers corresponding to the $(1,1)$-th element of matrices $A(t)$ and $B(t)$ for all $t$. Notice that this creates matrices $A'(t)$ and $B'(t)$ which have 0 as the $(1,1)$-th entry.

Under the transformation in Definition 4.4, the vector fields $\dot{A}'(t) = F(A')$ and $\dot{B}'(t) = F(B')$ are given by:

$$\begin{aligned}
\dot{A}'(t) &= \Phi(\sigma(t)) - (v^\dagger\Phi(\sigma(t))v)\mathbb{1} \\
\dot{B}'(t) &= -\Phi^\dagger(\rho(t)) + (v^\dagger\Phi^\dagger(\rho(t))v)\mathbb{1}
\end{aligned} \tag{20}$$

where $\frac{d}{dt}\left(v^\dagger A(t)v\right)$ is given by $v^\dagger \frac{dA(t)}{dt}v$.

Moreover, the values of $\rho'(t)$ and $\sigma'(t)$ are defined as:

$$\begin{aligned}
\rho'(t) &= \exp(A'(t))/\mathrm{Tr}(\exp(A'(t))) \\
\sigma'(t) &= \exp(B'(t))/\mathrm{Tr}(\exp(B'(t)))
\end{aligned} \tag{21}$$

**Proposition 4.5.** *The dynamics of $\rho(t)$ and $\sigma(t)$ remain the same after undergoing the canonical transformation. Equivalently, $A'(t)$ and $A(t)$ (resp. $B'(t)$ and $B(t)$) admit the same strategy $\rho(t)$ (resp. $\sigma(t)$).*

**Proposition 4.6.** *The mappings $A'(t)$ and $\rho(t)$ (resp. $B'(t)$ and $\sigma(t)$) are diffeomorphic to one another.*

Proposition 4.6 will be of crucial importance to our proof technique, since we first prove recurrence for the system described by $A'(t)$ and $B'(t)$, then recover recurrence in $\rho(t)$ and $\sigma(t)$.

To show Poincaré recurrence of Equations 17, we first prove two key properties: (i) the flow of $\dot{A}'$ is volume preserving, meaning that the trace of the Jacobian of the respective vector fields $\dot{A}'(t) = F(A')$ and $\dot{B}'(t) = F(B')$ are zero, and (ii) $A'$ and $B'$ have bounded orbits from any interior initial condition. Then, Poincaré recurrence of $A'$ and $B'$ follows from Poincaré's recurrence theorem.

**Volume Conservation.** We introduce a lemma which shows that in two-player zero-sum quantum replicator dynamics, the canonical transformation produces a dynamical system which preserves volume.

**Lemma 4.7.** *For two-player zero-sum quantum replicator dynamics, the vector fields that arise as a result of the canonical transformation in Definition 4.4 are volume preserving.*

The proof of Lemma 4.7 follows from considering the flows of $A'(t)$ and $B'(t)$ and showing that the divergences of the vector fields defined in Equations 20 are equal to zero. A straightforward application of Liouville's theorem completes the proof.

**Bounded Orbits.** We now show that the transformed dynamical system always has bounded orbits when initialized on the interior of the space of probability density matrices.

**Lemma 4.8.** *For any finite initial points $A(0)$ and $B(0)$, the dynamics mapped to $A'(t)$ and $B'(t)$ via the transformation in Definition 4.4 have bounded orbits.*

The proof of Lemma 4.8 relies on Theorem 4.2, and leverages the hermicity of the matrices involved. Moreover, we use the fact that the canonically transformed matrices $A'(t)$ and $B'(t)$ have zero as the $(1,1)$-th element to bound the eigenvalues of $A'(t)$ and $B'(t)$ away from infinity.

Now we are ready to prove Theorem 4.3 using Lemmas 4.7 and 4.8.

*Proof of Theorem 4.3.* By Lemmas 4.7 and 4.8, as well as the Poincaré recurrence theorem introduced in Section 2, we immediately see that the system of replicator equations given by $dA'/dt$ and $dB'/dt$ are Poincaré recurrent since they are volume preserving and have bounded orbits. Since the flows of $A'(t)$ and $\rho(t)$ are diffeomorphic to one another (likewise for $B'(t)$ and $\sigma(t)$), $d\rho/dt$ and $d\sigma/dt$ are also Poincaré recurrent. This concludes the proof. $\qquad\square$

All proofs of the results in this section are provided in Appendix D.

## 5 Experimental Results

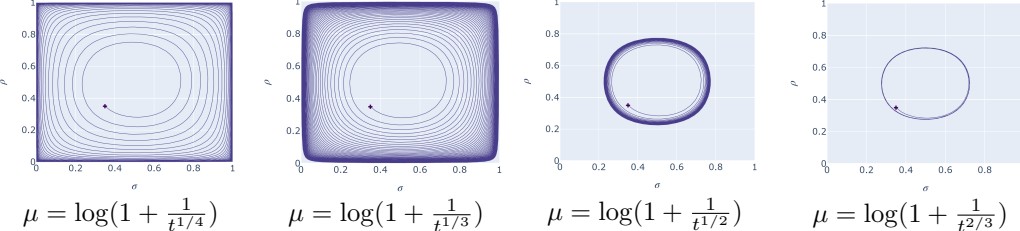

$$\mu = \log(1 + \tfrac{1}{t^{1/4}}) \qquad \mu = \log(1 + \tfrac{1}{t^{1/3}}) \qquad \mu = \log(1 + \tfrac{1}{t^{1/2}}) \qquad \mu = \log(1 + \tfrac{1}{t^{2/3}})$$

Figure 1: Eigenvalue trajectories for quantum Matching Pennies game with decreasing $\mu$ values.

To corroborate the theoretical results presented in prior sections, we performed relevant simulations of quantum games using both discrete MMWU and replicator dynamics. In the rest of this section, we standardize the use of quantum game matrices obtained via basis transform (described in more detail in Appendix E). This effectively allows us to transform classical games to the matrix setting.

First, we show the trajectories of the first eigenvalue of each player in a quantum Matching Pennies game, obtained using the discrete MMWU algorithm. We see that in accordance to Theorem 3.5,

the rate of divergence of the trajectories from the uniform Nash goes to zero for cases with rapidly decreasing learning rate $\mu$.

In the case of replicator dynamics, we present Bloch sphere representations of the trajectories in a quantum Matching Pennies game. The Bloch sphere is a unit 2-sphere representation of a qubit, and we utilize it to visualize the orbits of the replicator dynamics. In particular, the density matrix representing the strategy of each player at each time-step is given as a point within the sphere, and we plot the movement of these orbits over time. According to Theorems 4.2 and 4.3, we expect the trajectories of the replicator dynamics to stay on the interior of the Bloch sphere, since the surface of the sphere corresponds to the pure states of the system. We see from Figure 2 that over time, the system never reaches the boundary of the sphere, which experimentally agrees with our theory.

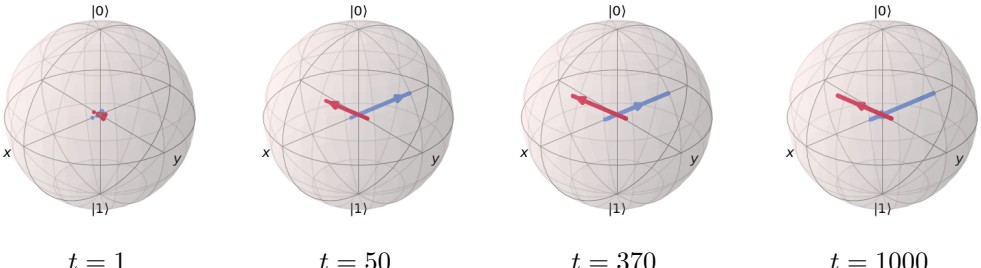

$$t = 1 \qquad\qquad t = 50 \qquad\qquad t = 370 \qquad\qquad t = 1000$$

Figure 2: Bloch sphere trajectories for quantum Matching Pennies game between Alice (blue) and Bob (red). The arrowheads represent the current state of each player at each time-step. Notice that over time, the orbits oscillate within the interior of the Bloch sphere.

Finally, in the experiments above we have only studied single qubit systems. We provide further experiments which show that our results hold beyond the single qubit setting in Appendix E.2.

## 6 Conclusion

In this paper, we studied the properties of Matrix Multiplicative Weights Update and its continuous analogue, quantum replicator dynamics, in the context of two-player zero-sum quantum games. First, we provide a formulation of quantum replicator dynamics which arises from MMWU. Then, we show that such systems exhibit quantum information-theoretic constant of motions. Finally, we show that in quantum replicator systems with interior Nash equilibria, the dynamics exhibit Poincaré recurrence.

This work constitutes an initial step towards analyzing learning behaviour in games with quantum information. In the classical world, showing that conservation laws and recurrence holds has led to a better understanding of game dynamics in increasingly complex settings [52, 43, 47]. Similar work is now potentially possible in the quantum setting.

Moreover, one key implication which can be derived from our recurrence results is to encourage novel discretization methods that preserve the connections to volume preservation and more generally, conservative dynamical systems (e.g. Hamiltonians). This line of research has received increased interest in recent years [57, 56, 17]. It is a natural question to explore whether such techniques can lead to similar advantages in the quantum setting as well.

Finally, some other interesting directions for future work include:

- extending our results to multi-agent network generalizations of quantum zero-sum games,
- understanding the potential behavior of quantum GANs using our results about learning in quantum zero-sum games, and
- understanding the quantum setting for different classes of games, e.g., potential games.

**Acknowledgements**

This research/project is supported by the National Research Foundation, Singapore and DSO National Laboratories under the AI Singapore Programme (AISG Award No: AISG2-RP-2020-016), NRF2019-NRFANR095 ALIAS grant, grant PIE-SGP-AI-2020-01, NRF 2018 Fellowship NRF-NRFF2018-07

and AME Programmatic Fund (Grant No. A20H6b0151) from the Agency for Science, Technology and Research (A*STAR). Rahul Jain's research is supported by the National Research Foundation, Singapore, also through the grant NRF2021-QEP2-02-P05, and the Ministry of Education, Singapore under the Research Centers of Excellence program. Ryann Sim gratefully acknowledges support from the SUTD President's Graduate Fellowship (SUTD-PGF).

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
