# OpenReview forum: "Matrix Multiplicative Weights Updates in Quantum Zero-Sum Games: Conservation Laws & Recurrence"
_NeurIPS.cc/2022/Conference — NeurIPS 2022 Accept_

### Official Review · Reviewer_TGZ4 · 2022-06-15

**Rating:** 7
**Confidence:** 5
**Soundness:** 4 excellent
**Presentation:** 4 excellent
**Contribution:** 3 good

**Summary:**

Quantum computing is an interesting research area these days, and this paper studies a specific problem in quantum computing, quantum zero-sum games. The authors consider a standard algorithm in zero-sum games, matrix multiplicative weight update (MMWU), and study its behavior of convergence in continuous time, in the sense of dynamical systems. Specifically, the authors prove that under the quantum replicator dynamics, the system exhibits conservation laws in a quantum information-theoretic sense. Moreover, the system exhibits the Poincare recurrence, i.e., all orbits return arbitrarily close to their initial conditions infinitely often. Numerical experiments are performed to corroborate the theoretical claims.

**Questions:**

My question is about the connection between this work and the machine learning community in general. Please see my discussions above.

**Limitations:**

Although the checklist mentioned that the limitations of the paper are described in the introduction and in the main text, I found a bit hard to locate that explicitly... It would be helpful if the authors can highlight this.

**Strengths And Weaknesses:**

In general, both quantum computing and game theory are very important and interesting fields, and it’s great to see that the authors are able to prove some theoretical results for quantum games. As far as I see, previous quantum computing papers relevant to the topic studies different perspectives, and this result on the conservation law of quantum games under MMWU is novel to me. This is also a significant result and I believe that it can bring further attention to the interdisciplinary research between quantum computing and game theory.

In all, I think this paper can appropriately fit into NeurIPS 2022. However, the paper still has space to improve. In particular, the authors might be able to make more efforts to build the relevance of this work to the general machine learning audiences. To me, the conservation law & recurrence result is definitely nice to know, specifically that the authors are able to prove this in the quantum computing domain, but these are more like pure math results and I didn’t immediately see the implication to machine learning. From the perspective of game theory, does that imply any fast quantum/classical algorithm based on MMWU such that we can solve relevant quantum/classical game theory problems (or maybe other problems relevant to machine learning) faster?

For the specific perspective of dynamical systems which covers the conservation law and recurrence properties studied by the paper, a similar story that has been well received by the ML community is the study of gradient descent methods in continuous time limit, in other words, characterization of optimization methods by differential equations. For instance, speaking of the methodology, this paper by Wibisono, Wilson, and Jordan published at PNAS https://www.pnas.org/content/113/47/E7351.short proposes a framework of continuous-time dynamics for understanding accelerated gradient descent methods, which is also based on the Bregman divergence (similar to this submission). It is noted that papers along this line have been well-received by the ML community with various NeurIPS and other top-venue papers. After reading this submission, I believe that it also has the potential to become more relevant as a systematic tool to understand game theory, but the authors should put more effort on this.

---

> ### Author Response · Authors · 2022-08-02
> **Response to Reviewer TGZ4**
>
> Thank you for your detailed review and support of our paper. In terms of interesting connections to ML, one key implication which can be derived from our conservation/recurrence results is to encourage novel discretization methods that preserve the connections to volume preservation and more generally conservative dynamical systems (e.g. Hamiltonians). In close connection to your comment, Wibisono, Tao and Piliouras [1] very recently developed and analyzed a discrete-time learning dynamic inspired by symplectic integrators which in classical zero-sum games performs very close to the continuous-time dynamics and has improved performance guarantees. It would be a natural question to explore whether such techniques provide similar advantages in the quantum setting as well. Moreover, as you point out such ideas have been received well by the ML community and we believe that our works adds an interesting contribution along these lines.
>
> [1] Wibisono, Andre, Molei Tao, and Georgios Piliouras. "Alternating Mirror Descent for Constrained Min-Max Games." arXiv preprint arXiv:2206.04160 (2022).

---

> > ### Comment · Reviewer_TGZ4 · 2022-08-03
> > **My thoughts after rebuttal**
> >
> > I would like to thank the authors for the reply. The paper by Wibisono, Tao and Piliouras is indeed interesting, and it would be nice if the future version of this paper can elaborate more along this line.

---

> > > ### Author Response · Authors · 2022-08-07
> > > **Response**
> > >
> > > Thank you for your response! We will more than happy to add a note in our relationship to this very recent work. In an nutshell, that paper shows that alternating Multiplicative Weights Updates performs much closer to its continuous-time "ideal" implementation in classical zero-sum games than previously studied algorithms and thus raises an interesting question whether a similar statement would hold in our case for alternating MMWU in quantum zero-sum games.  We believe that this is a rather interesting direction for future work and showcases how our ideas/settings are connected in interesting ways to other concurrently explored themes in optimization and learning in games.

---

### Official Review · Reviewer_VXhb · 2022-07-14

**Rating:** 5
**Confidence:** 3
**Soundness:** 3 good
**Presentation:** 3 good
**Contribution:** 3 good

**Summary:**

The paper studies a specific case of two-player zero-sum games where each player strategy is to probabilistically prepare a quantum pure state that he/she sends to a referee who then performs a joint measurement on the two quantum states to determine the payoffs of the players. The restriction is that each player updates their strategy according to the Matrix Multiplicative Weight Update (MMWU) methods. The validity of their strategies is guaranteed by the trace normalization performed after updating according to MMWU. The main results of the paper are Theorem 3.5 and 4.2: the total quantum relative entropy is a constant of motion.


**Questions:**

1. How are the main results similar or different from classical counterparts?
2. What happens if the two players share randomness and/or entangled states?
3. Will the experiments with multi-qubit strategies differ from the current experiments with single-qubit strategies?


**Limitations:**

Not applicable.

**Strengths And Weaknesses:**

I find the results of the papers are interesting with the strengths in the rigorous theoretical analysis deriving the main results of the papers.
Nevertheless, I am having difficulties to understand the impact of the results to the quantum learning theory. Especially, the results are restricted to the specific rules of updating strategies according to the MMWU, which seems to be quite restrictive.

Another weakness is the lack of comparison to the classical counterparts. Introducing quantum strategies can add more powers either to players. For example, what will happen if the two players share entangled states? Do Theorems 3.5 and 4.3 hold? Or, what if the players can only have shared random numbers but not entangled states?

I also think the experimental results are weak as they are essentially 1-qubit strategy for each player.

---

> ### Author Response · Authors · 2022-08-02
> **Response to Reviewer VXhb**
>
> Thank you for your review and comments. We would like to clarify that our key contribution in this paper is to establish that MMWU when applied to quantum zero-sum games exhibits Poincare recurrence. This is a very specific type of ``cycling” behavior that is typically exhibited in physical systems with conserved properties. Hence, the summary the reviewer has provided does not fully capture the nature of our contributions.
>
> Moreover, while focusing the analysis to MMWU might seem restrictive from a broader learning theoretical perspective, this is not so. MMWU is one of the key matrix update rules which have been applied to many settings (see our related work section). MMWU is also an object of seminal importance for quantum computation as the seminal QIP=PSPACE breakthrough result was proved by applying a parallelized form of the MMWU to a class of semidefinite programs that captures the computational power of quantum interactive proofs. There is a wider class of algorithms known as Matrix FTRL, but we leave the analysis in these settings to future works, since MMWU already has many interesting applications in the ML/quantum space and is the prototypical element of this class (See [1,2] below and refs therein).
>
> To answer the reviewer’s questions, our results are both conceptually as well as technically significantly harder than any of their classical analogues. For example, the only other theoretical result that we know about learning in our setting of quantum games is that of Waltrous and Jain which study exactly our setting and use the time-average convergence result prove the containment of the complexity class QRG(1), i.e. one-turn quantum refereed games, wherein the referee is specified by a quantum circuit, in PSPACE. These ideas and particularly the use of MMWU led to the seminal result of QIP=PSPACE. Hence this setting is of particular interest and understanding its day-to-day behavior leads to deeper insights of a basic algorithm in the space.   To prove these results we need new tools and techniques such as the designed canonical transformation of the statespace using the variables A’(t) and using quantum relative entropy as an invariant energy function.
>
> Critically, our analyzed learning algorithm (MMWU) is not a quantum but a classical algorithm. This of course has the advantage that we can run simulations of the algorithm and verify/showcase its theoretically predicted behavior. On the other hand, it sets interesting benchmarks against other truly quantum learning algorithms that try to leverage shared/entangled states but this is clearly not the setting we are exploring here. Similarly to the classical zero-sum games where many different learning algorithms and their properties are studded e.g. extra-gradient methods, optimistic methods, consensus optimization, adaptive methods, etc one would expect a similar proliferation of different ML algorithms in quantum settings and we are happy to take a step in that direction. Understanding the behavior of classical ML algorithms in quantum games allows for the possibility of proving the supremacy of some future quantum ML technique, however, this is clearly well beyond the scope of the current paper.
>
> Finally, we primarily experimented with single qubit systems since they are easily visualized on the Bloch sphere, but numerically we have performed simulations on multi-qubit systems, and the recurrence property holds in those settings as well. We would be happy to include more simulations in the camera-ready version of the paper.
>
> [1] https://lucatrevisan.wordpress.com/2021/11/10/online-optimization-post-7-matrix-multiplicative-weights-update/
>
> [2] Allen-Zhu, Zeyuan, Zhenyu Liao, and Lorenzo Orecchia. "Spectral sparsification and regret minimization beyond matrix multiplicative updates." Proceedings of the forty-seventh annual ACM symposium on Theory of computing. 2015.

---

> ### Author Response · Authors · 2022-08-09
> **Updates on Experimental Results**
>
> Dear Reviewer VXhb,
>
> Thank you again for your review, and for bringing up concerns regarding the size of experiments run in the paper. We would like to update you that we have run additional experiments on 2 and 3-qubit systems, showcasing that even in more complicated (but still zero-sum) settings, the quantum replicator dynamics still showcase recurrent behavior. On a more granular scale, we have also observed that the intermediate result stating that the sum of quantum relative entropies between the players and the Nash equilibrium remains constant still holds in multi-qubit systems. This clearly shows that our results are scalable to higher dimensional systems than previously indicated in our experiments. We have added new plots and discussion regarding this point to the appendix in our current draft, and we hope that this can convince you to recommend our paper for acceptance. Please let us know if you have any other questions or concerns!

---

> > ### Comment · Reviewer_VXhb · 2022-08-09
> > **Thanks for updating the experiments!**
> >
> > I thank the authors for their efforts to update the experiments. Single-qubit experiments were too simple but as the results also hold for multi-qubit cases, I am convinced that the merits of accepting the paper outweight its rejection.

---

### Official Review · Reviewer_wK2r · 2022-07-15

**Rating:** 6
**Confidence:** 3
**Soundness:** 3 good
**Presentation:** 3 good
**Contribution:** 2 fair

**Summary:**

The authors consider (2 player) quantum zero-sum games and analyze two algorithms for learning the Nash equilibrium: matrix multiplicative weights and its continuous-time analog, the quantum replicator dynamics. As far as I understand, both algorithms have been previously described in the literature. The authors prove the convergence some entropic quantities during the execution of the algorithms. The main technical contribution is the proof of Poincare recurrence of quantum replicator dynamics.

**Questions:**

 - Why do you analyze the change in the relative entropy instead of its actual value? Naively, it seems that proving that the change converges to 0 implies only that the relative entropy has converged to a fixed value which might or might not be 0. Has [25] already estabilished this convergence?
 - Are Theorems 3.5 and 4.2 interesting in their own right? How are they used to prove the main result, Theorem 4.3?

**Limitations:**

The work is purely theoretical, no direct social impact.

**Strengths And Weaknesses:**

The paper is well-written and provides just enough details to be understood at a superficial level. On the more fundamental level, some motivation and intuition should be given about the connections between section 4.1 and the part of the paper before it. Some space could be easily reclaimed by removing the definitions and claims that are not used outside of the appendix.

---

> ### Author Response · Authors · 2022-08-02
> **Response to Reviewer wK2r**
>
> Thank you for your review and suggestions. We can certainly reorganize the preliminaries to make space for more motivation regarding the recurrence result. Furthermore, see below for answers to your specific questions:
>
> > Why do you analyze the change in the relative entropy instead of its actual value? Naively, it seems that proving that the change converges to 0 implies only that the relative entropy has converged to a fixed value which might or might not be 0. Has [25] already estabilished this convergence?
>
> Since the time-derivative of the relative entropy is zero, this implies that its value remains constant. It does not converge to any value, e.g. 0. Thinking of relative entropy as a notion of pseudodistance this statement says that the state of the system stays at a constant distance from Nash equilibrium. It does not converge to it, as in this case it would be necessary for relative entropy to decrease and converge to zero. It does not diverge to infinity, which implies that the state of the system stays well mixed. Specifically, since the Nash equilibrium is interior, the relative entropy would be infinite if and only if the kernel of the state of the system was non-empty (the PSD matrix has a zero-eigenvalue). By continuity of the relative entropy and invariance and thus boundedness of the sum of non-negative relative entropies, we derive that the states of both agents have to remain bounded away from such boundary states. I.e. all eigenvalues of the states of the agents are lower bounded by some positive constant.  In the proof of Lemma 4.8, in lines 693-694 in the supplementary we leverages this lower bound on the eigenvalues to show that the system remains bounded in the space of transformed A(t) variables. Critically, none of the above arguments appeared in [25]. [25] analyzes only the time average performance of MMWU. Here we analyze the last-iterate or day-to-day behavior that requires much more subtle arguments.
>
> > Are Theorems 3.5 and 4.2 interesting in their own right? How are they used to prove the main result, Theorem 4.3?
>
> Yes they are for multiple reasons. First, as stated above since quantum relative entropy is a notion of pseudo-distance to Nash (\geq 0, zero iff the two psd matrices are equal (Klein’s inequality), jointly convex in arguments) the fact that the summation of these two distances is constant at the continuous time limit implies that the only way one agent can get closer to equilibrium is for the other agent to move away from it and this seesawing movement continues forever in a recurrent fashion. Secondly, this is a rather non-trivial property that it is easy to depict experimentally even in the space of numerical instabilities (as we do in page 23 fig 3 of the supplementary). Moreover, the connection between ML and conserved quantities has received considerable attention recently. E.g. [39] in our paper but also [1] about connections to Hamiltonians and multiple follow-ups. For example, a natural way to force convergence in such conservative systems is to explicitly add friction that leads to a constant decrease of this pseudodistance until convergence. Such friction in Follow-the-Regularized leader dynamics scale well [2] and have very recently been applied successfully in solving the game of Stratego [3] despite its astronomically large state space. Hence these properties can also inspire novel algorithmic techniques.
>
> [1] Balduzzi, David, et al. "The mechanics of n-player differentiable games." International Conference on Machine Learning. PMLR, 2018.
>
> [2] Perolat, Julien, et al. "From Poincaré recurrence to convergence in imperfect information games: Finding equilibrium via regularization." International Conference on Machine Learning. PMLR, 2021.
>
> [3] Perolat, Julien, et al. "Mastering the Game of Stratego with Model-Free Multiagent Reinforcement Learning." arXiv preprint arXiv:2206.15378 (2022).

---

### Official Review · Reviewer_Tz9L · 2022-07-16

**Rating:** 7
**Confidence:** 4
**Soundness:** 3 good
**Presentation:** 3 good
**Contribution:** 3 good

**Summary:**

A two-player zero-sum quantum game consists of three steps.

1) Two players prepare two mixed states $\sigma,\rho$.
2) A third party makes a joint measurement $R$ on the tensor product $\rho\otimes \sigma$.
3) The payoff of the first player receives is the result of the measurement, whereas the second player receives the negative of that value.

One may naturally define a notion of Nash equilibrium for such games. The general problem addressed here is the following: assuming a repeated game, do both players converge to the/a Nash equilibrium?

The present paper considers this problem in the case where both players follow a matrix multiplicative weights (MMW) algorithm, which (for suitable step sizes) achieves vanishing regret. Before I state the main results of this submission, I wish to mention two results in the corresponding classical setting with the multiplicative weights (MW) algorithm.

1) Bailey and Piliouras (2018 ACM Conference on Economics and Computation) prove that (under suitable assumptions) there is no convergence to Nash equilibria.
2) Mertikopoulos et al (SODA 2018) that players' states are Poincaré recurrent in a continuous-time limit of MW (or more generally of "follow the regularized leader").

The present paper proves variants of the above results for the continuous-time limit of MMW, called quantum replicator dynamics (under suitable conditions). The proof of Poincaré recurrence is related to the second reference above, in that it requires showing that the quantum replicator flow is volume preserving (after a certain transformation) and has bounded trajectories. Another result shows that the quantum replicator dynamics leaves invariant a sum of relative entropies, each comparing "one half of a Nash equilibrium"(ie. one of the states in the pair constituting the equilibrium) to "one half of the quantum replicator dynamics". Additional bounds are given for the evolution of the relative entropies in discrete time.

The above results are complemented by experiments.The introduction motivates these results by their relationship to quantum learning and quantum GANs. These connections are taken up in the conclusion as part of a discussion on future directions.


**Questions:**

Q1) Can you explain the results about Quantum GANs that you would like to formalize (as per your conclusion)?

Q2) In the argument for Theorem 3.5, $\rho_j$ and $\sigma_j$ also depend on $\mu$. However, I do not quite see how the proof of the theorem accounts for this. Can you explain this point? Also, the reasoning in line 648 about evolving $\rho_j$ does not seem very rigorous (thought it could be made rigorous).

Q3) Can you explain how exactly Definition 4.4 is used in the proof? Where would the proof "break" if this definition did not hold?

*Update on 08/07*

I thank the authors for their clarifications.

**Limitations:**

The submission is good in this regard. It is "limited" in that it does not fully generalize the results from the classical setting.

**Strengths And Weaknesses:**

*Strengths*

The paper is mostly clearly written (although not everywhere). Its main strength of the present paper is the (partial) extension of the two papers mentioned in the summary to the quantum setting. The conservation of the sum of relative entropies is a nice result, and the proof, although relatively simple, is different from those in the main references.

*Weaknesses*

1. Unlike the first paper papers in the above summary, the present submission only considers the continuous-time limit, not the actual repeated game.
2. Unlike the second paper, only multiplicative weights are considered (though this may be a limitation of online learning theory in the matrix setting).
3. Although the main results are motivated by applications to quantum GANs, the connection seems somewhat tenuous. This raises the question of whether NeurIPS is a good fit for the paper.
4. I have some reservations about the proofs of Theorems 3.5 and 4.3. Mostly they stem from the fact that $\rho_j,\sigma_j$ also depend on $\mu$. In particular, there seems to be a "order of limits" question there that I will discuss later.

--

*Other comments \& suggested corrections*

Line 153: "can also be defined"- no, that's the actual definition.

Line 158: in general, how does one define a differential equation in a topological space?

The notation $A(t)$ used in Liouville's Theorem has another meaning later in the paper.

The use of time derivatives in (17) is confusing.

Line 234: here "typically" apparently should mean "in the classical (or commuting) case".

--

*Review update on 08/07*

The authors' answers to point 4 above have convinced me. I am satisfied that the proofs are correct. I am also clearer on the literature and their motivation for considering this specific algorithm.

---

> ### Author Response · Authors · 2022-08-02
> **Response to Reviewer Tz9L (Part 1/2)**
>
> Thank you for your detailed review and feedback. Please see below for our responses to your questions:
>
> > Unlike the first paper papers in the above summary, the present submission only considers the continuous-time limit, not the actual repeated game. Unlike the second paper, only multiplicative weights are considered (though this may be a limitation of online learning theory in the matrix setting).
>
> The reviewer here is missing a critical reference: [Piliouras and Shamma SODA’14]. This paper is the critical precursor to both cited papers [Bailey and Piliouras EC‘18; Mertikopoulos et al SODA‘18] and in fact addresses exactly the classical analogue of our setting: MW in continuous-time (i.e. replicator) in classical zero-sum games showing recurrence. Hence this clearly shows that our setting is important enough for stand alone investigation. Similarly, to the ‘14 paper we believe that our result will allow for analogous interesting extensions for discrete-time and different regularizers but this is beyond the scope of the current work. Notably even in the classical case it took 4 years for the follow-up results to appear in the literature showing that these are nontrivial generalizations.
>
> Piliouras, Georgios, and Jeff S. Shamma. "Optimization despite chaos: Convex relaxations to complex limit sets via Poincaré recurrence." Proceedings of the twenty-fifth annual ACM-SIAM symposium on Discrete algorithms. Society for Industrial and Applied Mathematics, 2014.
>
> > Although the main results are motivated by applications to quantum GANs, the connection seems somewhat tenuous. This raises the question of whether NeurIPS is a good fit for the paper.
>
> Although our theoretical model is indeed loosely inspired by quantum GANs, one can also point out many recent papers in ML conferences in classical zero-sum games that similarly have only loose connections to the actual technicalities of GANs (e.g. bilinear games, or convex-concave games instead, not taking into account that GANs output distributions/measures over complex spaces instead of single points e.t.c.).
>
> More importantly, one can motivate our setting from numerous perspectives, e.g., its minimality. If the quantum setting is interesting for ML, then our setting is the simplest competitive quantum ML setting and thus a natural starting point for formal investigation. From a third perspective, it is also a natural generalization of learning in classical zero-sum games that have recently been the object of intense investigation at NeurIPS and related conferences. E.g. In NeurIPS last year [1] established Poincare recurrence for continuous time no-regret in evolving bilinear zero-sum games. [2] was a spotlight paper in NeurIPS’20 studying continuous-time no-regret dynamics such as replicator/MWU in normal form games showing that they do not converge to mixed NE. [3] was another NeurIPS paper establishing Poincare recurrence for continuous-time learning dynamics in another setting of zero-sum games. [4] is another well cited NeurIPS paper that studies learning dynamics in zero-sum games from a dynamical systems perspective. So, we believe that the subject matter and the nature of results are a great match for NeurIPS.
>
> [1] Fiez, Tanner, et al. "Online Learning in Periodic Zero-Sum Games." Advances in Neural Information Processing Systems 34 (2021): 10313-10325.
>
> [2] Vlatakis-Gkaragkounis, et al. "No-regret learning and mixed nash equilibria: They do not mix." Advances in Neural Information Processing Systems 33 (2020): 1380-1391.
>
> [3] Vlatakis-Gkaragkounis, et al. "Poincaré recurrence, cycles and spurious equilibria in gradient-descent-ascent for non-convex non-concave zero-sum games." Advances in Neural Information Processing Systems 32 (2019).
>
> [4] Daskalakis, Constantinos, and Ioannis Panageas. "The limit points of (optimistic) gradient descent in min-max optimization." Advances in neural information processing systems 31 (2018).
>
> > Can you explain the results about Quantum GANs that you would like to formalize (as per your conclusion)?
>
> We would like to clarify that in the conclusion, we simply mean that we would like to study the theoretical properties of QGANs in much the same way that classical GANs have been studied using tools from game theory. For instance, the observation of limit cycling behavior in classical GANs experiments has been explained via the theoretical analysis of game dynamics. It is not unreasonable to expect similar behavior in QGANs, which our work would explain. We are happy to rephrase that sentence for the sake of clarity.

---

> ### Author Response · Authors · 2022-08-02
> **Response to Reviewer Tz9L (Part 2/2)**
>
> > In the argument for Theorem 3.5, ρj and σj also depend on μ. However, I do not quite see how the proof of the theorem accounts for this. Can you explain this point? Also, the reasoning in line 648 about evolving ρj does not seem very rigorous (thought it could be made rigorous).
>
> Consider the MMWU update from time j-1 to j. As $\mu \to 0$, we claim that $\rho_j$ and $\sigma_j$ do not change more than $O(\mu)$. Indeed, since all payoffs in the game are bounded in the MMWU update from time j-1 to j, all entries in the numerators increase by at most $exp(O(\mu)) = 1 + O(\mu)$. Likewise, the denominator is at least as large, but also upper bounded by the previous value of the denominator*$(1+O(\mu))$. Hence, every entry in the outputs $\rho_j$ and $\sigma_j$ is at most $O(\mu)$ from $\rho_{j-1}$ and $\sigma_{j-1}$. Moreover, we have that $Tr (\rho_j \Phi(\sigma_{j−1}))$, $Tr (\rho_{j-1}\Phi(\sigma_j))$ and $Tr (\rho_{j-1}\Phi(\sigma_{j−1}))$ are all within O(\mu) of each other. The rest of the proof follows as written.
> Regarding the reasoning in line 648, if we perform the substitution $j\to t$, $j-1 \to t-dt$ and $\mu \to dt$, we obtain a discrete-time setting with stepsizes of duration $\mu$. As $\mu \to 0$, then, taking the limit of the sum of quantum relative entropies in this discrete setting results is by definition the time derivative of the sum of quantum relative entropies. This derivative by Theorem 3.5 is equal to zero.
> We will be sure to clarify these points in more detail in the camera ready version of the paper.
>
> > Can you explain how exactly Definition 4.4 is used in the proof? Where would the proof "break" if this definition did not hold?
>
> In order to prove that recurrence holds in this setting we need to show that a) volume is conserved, and b) the orbits of the system remain bounded away from the boundary. The first part is relatively straightforward and follows by applying Liouville’s theorem, but proving bounded orbits is more challenging. Indeed, the reason for the canonical transformation is to design a diffeomorphic system to the original dynamical system where proving boundedness is simpler. This is because by construction we are able to obtain bounds on the maximum and minimum eigenvalues of $A’(t)$, which leads eventually to the conclusion that the entries of $A(t)$ are bounded. One can give an analogy using intuition from classical game theory: The parameters $A(t)$ are effectively integrals of payoffs for different actions. Clearly, if all the payoffs entries of an agent are strictly negative/positive then the time integrals of these quantities will diverge to infinity. Hence we need to ``regularize” these variables in a way that they remain bounded while preserving exactly the necessary information to compute the strategies of the agents. This novel transformation that we have introduced satisfies both these criteria. This is indeed a subtle point in the analysis and we will be happy to expand on the intuition behind this technique.

---

> > ### Comment · Reviewer_Tz9L · 2022-08-08
> > **Thank you; score updated**
> >
> > I thank the authors for their detailed response, which clarified several important points. I am now of the opinion that the paper should be accepted, and my score has been updated accordingly.

---

> > > ### Author Response · Authors · 2022-08-09
> > > **Thank you!**
> > >
> > > Thank you for your quick response to our rebuttal and for your support!

---

### Author Response · Authors · 2022-08-07
**Thank you & welcoming more discussion.**

Dear Reviewers,

Thank you very much again for your time for reviewing our paper and for your helpful comments and suggestions. We are particularly grateful to Rev. TGZ4 for their engaging discussion and response to our comments. We were wondering if there is anything else the rest of the reviewers would like to discuss. We would very much like to engage with you in our responses to your suggestions. If you have any remaining questions, please feel free to post them here, and we would be more than happy to discuss them further with you.

Best Regards,
the Authors.

---

### Meta-Review · Area_Chair_uico · 2022-08-26

**Recommendation:** Accept
**Confidence:** Less certain

**Metareview:**

In this submission, the authors consider zero-sum games and analyze two algorithms for learning the Nash equilibrium. In this version, each player's strategy is to probabilistically prepare a "quantum pure state" that is sent to a referee who performs a joint measurement on the two quantum states to determine the payoffs of the players. This is an interesting generalization of zero-sum games to quantum computing.

While both algorithms (matrix multiplicative weights and its continuous-time analog, the quantum replicator dynamics) have been previously described in the literature. The authors prove many interesting new results, including the convergence of some observable, and the Poincare recurrence of quantum replicator dynamics.

The consensus among reviewers is that the paper is clearly written, and definitely presents a significant extension of known results on such problems. Following the reviewers' suggestions, the authors have expended their discussion on the relevance of this work to the general machine learning audiences.

**Award:**

No

---

### Decision · Program_Chairs · 2022-09-14

Accept